# MA4DIV: Multi-Agent Reinforcement Learning for Search Result Diversification

## Abstract

Search result diversification (SRD), which aims to ensure that documents in a ranking list cover a broad range of subtopics, is a significant and widely studied problem in Information Retrieval and Web Search. Existing methods primarily utilize a paradigm of "greedy selection", i.e., selecting one document with the highest diversity score at a time or optimize an approximation of the objective function. These approaches tend to be inefficient and are easily trapped in a suboptimal state. To address these challenges, we introduce **M**ulti-**A**gent reinforcement learning (MARL) for search result **DIV**ersity, which called **MA4DIV** [1]. In this approach, each document is an agent and the search result diversification is modeled as a cooperative task among multiple agents. By modeling the SRD ranking problem as a cooperative MARL problem, this approach allows for directly optimizing the diversity metrics, such as $\alpha$-NDCG, while achieving high training efficiency. We conducted experiments on public TREC datasets and a larger scale dataset in the industrial setting. The experiemnts show that MA4DIV achieves substantial improvements in both effectiveness and efficiency than existing baselines, especially on the industrial dataset.

## CCS Concepts

• **Information systems → Information retrieval diversity**.

## Keywords

Search Result Diversification; Learning to Rank; Multi-Agent Co-operation; Reinforcement Learning

**ACM Reference Format:**

Anonymous Author(s). 2018. MA4DIV: Multi-Agent Reinforcement Learning for Search Result Diversification. In *Proceedings of Make sure to enter the correct conference title from your rights confirmation emai (Conference acronym 'XX)*. ACM, New York, NY, USA, 13 pages. https://doi.org/XXXXXXX.XXXXXXX

## 1 Introduction

How to provide diverse search results to users is an important yet complicated problem in information retrieval. The search result diversification (SRD) [1] means that for a user's search query, we need to provide search results that cover a wide range of subtopics to meet the requirements of different users who may have multiple

---

[1]Anonymous code can be seen on https://anonymous.4open.science/r/MA4DIV-C2A8.

interpretations or intentions. Many diversity-aware metrics, such as $\alpha$-NDCG [7], ERR-IA [4], S-recall [49], have been proposed to evaluate the effectiveness of search result diversification. Since the diversity of a given document is affected by the documents that precede it, optimizing some of the diversity metrics is NP-Hard [3]. Therefore, search result diversification has become an important and challenging research topic.

Existing works on SRD can primarily be categorized into three main approaches:

**Greedy Selection Approaches** The typical search result diversification methods use "greedy selection" to sequentially select documents to build a diversified ranking list in a step-by-step manner. At each step, the ranking model selects the best document for the current ranking position according to the additional utility it can bring to the whole ranking list. Different methods use different criteria in the greedy selection. Maximal marginal relevance (MMR) [2] algorithm proposes maximal marginal, and takes the maximum document distance as the utility. xQuAD [28] is another widely used diverse ranking model which defines the utility to explicitly account for relationship between documents and the possible subqueries. In recent years, many methods based on machine learning [19, 21, 25, 44, 54] have been proposed to solve the search result diversification task.

**Single-Agent Reinforcement Learning Approaches** While the greedy selection approaches only consider the myopic utility brought by each document, RL approaches can better models the current state (i.e., the documents examined by the user) and future expected reward. Xia et al. [42] propose the MDP-DIV framework to model the search result diversification ranking as Markov Decision Process (MDP) [23], and uses a reinforcement learning algorithm [38] to optimize the ranking model. While MDP-DIV still use a "greedy selection" to select next candidate result, it can easily lead to sub-optimal ranking list. $M^2$Div [11] introduces Monte Carlo Tree Search (MCTS) [8] for high-quality exploration. This method helps to alleviate the problem of being stuck in a sub-optimal solution at a cost of increasing training and inference time.

**Simultaneous Scoring and Ranking Approaches** Some recent studies propose to simultaneously score all candidate documents and obtain the whole ranking list by sorting these scores in a descending order. This paradigm can greatly improve the inference efficiency as the documents can be scored in parallel. In particular, DALETOR [45] and MO4SRD [48] propose a differentiable proxy for the non-differentiable diversity metric $\alpha$-NDCG [7], which enables a gradient-based optimization for the diversity metric. Other methods [9, 17, 24, 30, 31] optimize the diversity of ranking list based on list-pairwise loss fuction [39]. However, although both the differentiable approximation of the optimization objective and the list-pairwise loss function are good approximations to the true optimization objective, such as alpha-NDCG, they are still not equal to the true optimization objective. Therefore, the solution may still

be biased and sub-optimal. In Appendix A, we further explain why existing methods lead to suboptimal diversity ranking results.

To summarize, although the problem of search result diversification has been extensively investigated, existing methods still have some limitations in terms of the effectiveness and efficiency. From the perspective of effectiveness, the Greedy Selection Approaches (e.g. [2, 28]) and the Simultaneous Scoring and Ranking Approaches (e.g [17, 31, 45, 48]) may be sub-optimal in optimizing the end diversity metric, such as $\alpha$-NDCG and ERR-IA. From the perspective of efficiency, those Single-Agent Reinforcement Learning Approaches face a intrinsic challenge of exploring a huge space of all possible rankings, which inevitably leads to sub-optimal solution [42] and high time-complexity in both training and inference [11].

In this paper, in order to alleviate the problems of existing methods mentioned above, we propose to formalize the diverse ranking as a multi-agent cooperation process. Since **M**ulti-**A**gent reinforcement learning (MARL) algorithm is used to optimize the **DIV**ersity ranking model, we call this new method **MA4DIV**. Specifically, by treating each document as an independent agent within a cooperative multi-agent setting, we simulate a fully cooperative multi-agent task. In this setup, each agent (document) makes action selections based on observations that include the features of both the query and the documents, aiming to maximize a shared reward function that is directly related to the diversity evaluation metrics of search results. Furthermore, MA4DIV employs value decomposition [27] to optimize global diversity directly by the structures of mixing network and hypernetworks during training.

The advantages of our MA4DIV over existing works include:

- Compared to "greedy selection" paradigm, MA4DIV simultaneously predicts the ranking scores of all documents, which can improve the efficiency of ranking process.
- The ranking scores of all documents and a ranked documents list with diversity can be obtained in one time step. Therefore, MA4DIV has higher exploration efficiency than Single-Agent Reinforcement Learning algorithms, such as MDP-DIV, which must go through an entire episode to explore a diversity ranking list.
- The training of the MA4DIV model does not require any approximations of different diversity metrics, as it can directly optimize the diversity metrics as rewards during multi-agent reinforcement learning process.

In order to demonstrate the effectiveness and efficiency of MA4DIV, we conducted experiments on TREC benchmark datasets and a new industrial dataset. The experimental results on TREC datasets showed that, MA4DIV achieved the state-of-the-art performance on some evaluation metrics, and training time was significantly shorter than baselines. Considering the small number of queries in TREC dataset (only 198 valid queries), we built a larger diversity dataset based on the real search engine data, which is called DU-DIV. On the DU-DIV dataset, MA4DIV achieved the state-of-the-art performance on all evaluation metrics, and demonstrated a substantial improvement in exploration efficiency and training efficiency.

## 2 Related Works

### 2.1 Search Result Diversification

Carbonell et al. [2] introduced the maximal marginal relevance criterion, which uses a linear combination of query-document relevance and document novelty to determine the document to be selected. An extension of this concept, the probabilistic latent MMR model [13] was proposed by Guo and Sanner. Approaches such as xQuAD [28] directly model different aspects of a query, estimating utility based on the relevance of retrieved documents to identified sub-queries or aspects. Hu et al. [16] proposed a utility function that leverages hierarchical intents of queries, aiming to maximize diversity in the hierarchical structure. Other researchers, like He et al. [14], have proposed combining implicit and explicit topic representations to create better diverse rankings, while Gollapudi et al. [12] proposed an axiomatic approach to result diversification.

Machine learning techniques have also been applied to construct diverse ranking models, often adhering to the sequential document selection or greedy sequential decision-making framework. Several researchers have defined utility as a linear combination of hand-crafted relevance and novelty features [40, 44, 54]. In [41], novelty can be modeled using deep learning models such as neural tensor networks. Radlinski et al. [26] propose to learn a diverse ranking of documents directly based on users' clicking behavior.

Reinforcement learning algorithms also are proposed to improve the effectiveness of "greedy selection" paradigm in search result diversification. MDP-DIV [42] consider the diverse ranking process as a Markov Decision Process (MDP). In MDP-DIV, agent select one document in one time step, so a ranked list can be obtained after an episode. And the episode reward can be any metrics, such as $\alpha$-DCG. While Feng et al. [11] think that "greedy selection" paradigm is easily lead to local optimum, so $M^2$Div is proposed to introduce Monte Carlo Tree Search (MCTS) into MDP ranking process. In this way, $M^2$Div does alleviate the problem that tend to fall into local optimum by conducting high-quality exploration, but also brings expensive training and inference costs.

There are some methods based on score-and-sort paradigm. DALETOR [45] and MO4SRD [48] are proposed to derive differentiable diversification-aware losses which are approximation of different diversity metrics, such as $\alpha$-NDCG. Other methods, such as DSSA [17], DESA [24], Graph4DIV [30], KEDIV [31], CL4DIV [9], use list-pairwise loss function [39] to optimize the diversity of the ranking list.

### 2.2 Reinforcement Learning for IR

Xu et al. [37] formulate a ranking process as an MDP and train the ranking model with a policy gradient algorithm of REINFORCE [38]. Singh et al. [29] propose PG Rank algorithm to enhance the fairness of ranking process with reinforcement learning methods. PPG [43] uses pairwise comparisons of two sampled document list with in a same query, which makes an unbiased and low variance policy gradient estimations. Yao et al. [46] model the interactions between search engine and users with a hierarchical Markov Decision Process, which improves the personalized search performance. RLIRank [53] implement a learning to rank (LTR) model for each iteration of the dynamic search process. Zou et al. [55] formulate the ranking process as a multi-agent Markov Decision Process, where

the interactions among documents are considered in the ranking process. However, Zou et al. [55] conduct the single-agent reinforcement learning algorithm REINFORCE to optimize the ranking model.

## 2.3 Multi-Agent Reinforcement Learning

Value decomposition [5, 6, 27, 32, 35, 36] and Actor-Critic [20, 47, 50–52] are two typical branches of multi-agent reinforcement learning (MARL). Among these, QMIX [27] is the algorithm that first obtains the global utility function by nonlinear combination of individual utility functions, which can assign the global reward to each agent implicitly and nonlinearly. In this paper, we conduct the framework of QMIX as a reference to optimize the diverse ranking process.

## 3 Background

### 3.1 Co-MARL

In this work, we consider the process of diversity ranking as a fully cooperative multi-agent reinforcement learning (Co-MARL) task, which is formally defined as a tuple $G = \langle S, \mathcal{A}, r, \mathcal{Z}, O, n, \gamma \rangle$. $s \in S$ is the state of the environment. Each agent $i \in \mathcal{G} \equiv \{1, \ldots, n\}$ chooses an action $a_i \in A$ which forms the joint action $\mathbf{a} \in \mathcal{A} \equiv A^n$. The reward function which is modeled as $r(s, \mathbf{a}) : S \times \mathcal{A}$ is shared by all agents and the discount factor is $\gamma \in [0, 1)$. In our diversified search task, it follows fully observable settings, where agents have access to the state. Instead, it samples observations $z \in \mathcal{Z}$ according to observation function $O(s, i) : S \times \mathcal{A} \to \mathcal{Z}$. In our algorithm, the joint policy $\boldsymbol{\pi}$ is based on action-value function $Q_{tot}^{\pi}(s_t, \mathbf{a}_t) = \mathbb{E}_{s_{t+1}:\infty, \mathbf{a}_{t+1}:\infty}[\sum_{k=0}^{\infty} \gamma^k r_{t+k} | s_t, \mathbf{a}_t]$. The final goal is to get the optimal action-value function $Q^*$.

### 3.2 General Format of Test Set for Diversified Search

Suppose there is a given query $\mathbf{q}$ which is associated with a set of candidate documents $\mathbf{D} = \{\mathbf{d_1}, \ldots, \mathbf{d_n}\} \in \mathcal{D}$, where query $\mathbf{q}$ and each document $\mathbf{d_i}$ are represented as $L$-dimensional preliminary representations, i.e., the $L$-dimensional vector given by the BERT model [10], and $\mathcal{D}$ is the set of all candidate documents. Our objective is to sort the documents in the candidate set $\mathcal{D}$ in such a way that the documents ranked higher cover as many subtopics as possible, thereby improving the diversity evaluation metrics.

To compute the diversity metrics introduced above, we need to collect a test collection with relevance labels at the subtopic level. Therefore, a test collection with $N$ labeled queries can be formulated as:

$$\{(q^k, D^k), J^k\}_{k=1}^N$$

where $J^k$ is a binary matrix with $n \times m$ dimensions. $J^k(i, l) = 1$ means that document $\mathbf{d_i}$ covers the $l$-th subtopic in the given query $q^k$ and $J^k(i, l) = 0$ otherwise.

## 4 The MA4DIV Model

In this section, we first describe how to formulate the search results diversification problem as a Co-MARL by introducing the basic elements in the MA4DIV framework. Then, we present the architecture of the MA4DIV framework, as shown in Figure 1, and elaborate

the implementation of the Agent Network, Ranking Process, and Mixing Network. Finally, we will give an algorithm to train the MA4DIV.

### 4.1 Essential Elements of MA4DIV

In Section 3.1, we define the process of diversity ranking as a Co-MARL. Next, we introduce the essential elements of Co-MARL specified to our MA4DIV:

**Agents $\mathcal{G}$**: We model each document $i \in \{1, \ldots, n\}$ as agent $i \in \mathcal{G} \equiv \{1, \ldots, n\}$. In the setting of multi-agent cooperation, each document implements the ranking process by cooperating with each other.

**State $S$**: State signifies global information and holds a pivotal role in multi-agent cooperation. It is through the utilization of state $S$ that effective collaboration between agents (documents) can be established, resulting in a diverse and high-quality ranking list. State is defined $s$ as:

$$s = \{\mathbf{q}, \mathbf{D}\} \tag{1}$$

From Equation (1) we can see that state $s$ contains information for the given query $\mathbf{q}$ and information for all documents $\mathbf{D} = \{\mathbf{d_1}, \ldots, \mathbf{d_n}\}$.

**Observation $O$**: Observation represents the information that each agent receives. Each agent makes decisions based on its respective observation $\boldsymbol{o_i}$. The definition of $\boldsymbol{o_i}$ is shown as Equation (2).

$$\boldsymbol{o_i} = \{\mathbf{q}, \mathbf{D}, \mathbf{d_i}\} \tag{2}$$

where $\mathbf{q}$ is embedding vector of the given query, $\mathbf{D} = \{\mathbf{d_1}, \ldots, \mathbf{d_n}\}$ represents the embedding vectors of all documents and $\mathbf{d_i}$ is embedding vector of document $i$ to make $\boldsymbol{o_i}$ be personalized to each agent.

**Actions $\mathcal{A}$**: The action space $a_i \in \mathcal{A}$ of each agent is corresponding to a set of integer ranking scores $s_i \in \{1, \ldots, |\mathcal{A}|\}$, $|\mathcal{A}|$ is the dimension of the action space. When making decisions, agent $i$ will select an action $a_i$ from $\mathcal{A}$. And the action $a_i$ corresponds to a ranking score $s_i$, which will serve as the basis for agent $i$ (document $i$) to assess the level of diversity to itself, and thus, be used in generating the final ranking list.

Upon the selection of joint-action $\mathbf{a} = \{a_1, \ldots, a_n\}$ by all agents based on joint-observation $\mathbf{o} = \{\boldsymbol{o_1}, \ldots, \boldsymbol{o_n}\}$, the ranking scores of candidate documents list $\mathbf{D} = \{\mathbf{d_1}, \ldots, \mathbf{d_n}\}$ can be obtained as **scores** $= \{s_1, \ldots, s_n\}$. By sorting all documents according to **scores**, a ranked document list by diversity level can be achieved, called $\mathbf{D_{ranked}}$.

**Reward $\mathcal{R}$**: The reward function should guide the parameter updates of the model to achieve diversity of ranked documents. In search result diversification, there are some evaluation metrics, such as $\alpha$-NDCG, are used to evaluate the diversity of a ranked document list. So it's natural to consider these metrics as reward functions. In this paper, we define the reward function as $\alpha$-NDCG@k:

$$\mathcal{R}(\mathbf{D_{ranked}}) = \alpha\text{-}NDCG@k \tag{3}$$

**Episode $\mathcal{E}$**: In typical MDP, an episode contains multiple decision steps. So the objective function is cumulative rewards in an episode with a discount factor $\gamma$, defined as:

$$G_t = \sum_{k=0}^{n-1-t} \gamma^k \mathcal{R}_{t+k+1} \qquad (4)$$

where $n$ is the step number in an episode.

However, the episode in MA4DIV contains only one step. Since the complete ranked document list can be obtained in one time step, the reward can then be calculated for end-to-end training. Therefore, instead of implementing a multi-step decision through a state transition function, an episode can contain only one step. And the final objective function of MA4DIV can be written as:

$$G = \mathcal{R} = \alpha\text{-}NDCG@k \qquad (5)$$

## 4.2 Agent Network

In Figure 1, the green modules represent the Agent Network for an agent $i$, which share parameters with each other. On the left side of Figure 1 is the detailed structure of the Agent Network. It can be seen that we concatenate $\{\mathbf{q}, \mathbf{d_i}, \mathbf{e_i}\}$, which is precisely obtained from the observation for agent $i$ defined in Equation (2). And $\mathbf{e_i}$ is a vector with high-level cross feature between document $i$ and all other documents, computed by Multi-Head Self-Attention (MHSA) module. The detail of MHSA and the reason why we use MHSA can be found in Appendix C. Finally, the features of all documents $\mathbf{D} = \{\mathbf{d_1}, \ldots, \mathbf{d_n}\}$ is reshaped to output $\mathbf{e_i}$ as Equation (6):

$$\{\mathbf{e_1}, \ldots, \mathbf{e_i}, \ldots, \mathbf{e_n}\} = \mathbf{MHSA}(\{\mathbf{d_1}, \ldots, \mathbf{d_n}\}) \qquad (6)$$

Then, the action-values $Q_i^{a_m}, m \in \{1, \ldots, |\mathcal{A}|\}$ are calculated according to Equation (7). Specifically, $Q_i^{a_m} = Q(\boldsymbol{o_i}, a_m)$ represents the expected reward of taking a specific action $a_m$ in a given observation $\boldsymbol{o_i}$ for agent $i$, which is defined in Equation (8).

$$\{Q_i^{a_1}, \ldots, Q_i^{a_m}, \ldots, Q_i^{a_{|\mathcal{A}|}}\} = \mathbf{MLP}(\{\mathbf{q}, \mathbf{d_i}, \mathbf{e_i}\}) \qquad (7)$$

$$\begin{aligned} Q_i^{a_m} &= Q(\boldsymbol{o_i}, a_m) \\ &= \mathbb{E}[\mathcal{R}|\boldsymbol{o} = \boldsymbol{o_i}, a = a_m] \\ &= \mathbf{MLP}(\boldsymbol{o_i}) \quad \text{and } m \in \{1, \ldots, |\mathcal{A}|\} \end{aligned} \qquad (8)$$

In reinforcement learning (RL), a popular approach to select the action-value is the $\varepsilon$-greedy policy, which is used to balance exploration and exploitation during the learning process.

Specifically, with a probability of $1 - \varepsilon$, the agent $i$ chooses the action-value $Q_i^{a_*}$ which has the highest estimated value ($\max Q_i^{a_m}$) for the current observation $\boldsymbol{o_i}$, and the corresponding chosen action is $a_i^* = \arg\max_{a_m} Q_i^{a_m}, \forall m \in \{1, \ldots, |\mathcal{A}|\}$. This is the exploitation part, where the agent makes the best decision based on the current model. On the other hand, with a probability of $\varepsilon$, the agent $i$ chooses an action-value uniformly at random from $|\mathcal{A}|$ action-values. This is the exploration part, where the agents try to discover new permutations of the candidate documents set. Mathematically, this process can be written as Equation (9):

$$Q_i^{a_*} = \begin{cases} \max Q_i^{a_m}, \forall m \in \{1, \ldots, |\mathcal{A}|\} & \text{with } p{=}1{-}\varepsilon \\ \text{random } Q_i^{a_m}, \forall m \in \{1, \ldots, |\mathcal{A}|\} & \text{with } p{=}\varepsilon \end{cases} \qquad (9)$$

In the training process, the value of $\varepsilon$ is often initially set to a high value (e.g., 1.0), and gradually decayed to a low value (e.g.,

0.05) over training steps $t$ ($\varepsilon(t) = \max(0.05, 1 - \frac{t}{T})$), allowing the agent to explore the environment extensively at the beginning, and then exploit more as its knowledge increases. In the testing process, we set $\varepsilon = 0$ to get the optimal policy.

In this way, the $\varepsilon$-greedy policy used in MA4DIV helps to balance the trade-off between exploration and exploitation, which promotes to explore more different ranking permutation. And it's more conducive to update the parameters of the Agent Network.

## 4.3 Ranking Process

In the ranking process, the agents receive a query $\mathbf{q}$ and the associated documents $\mathbf{D} = \{\mathbf{d_1}, \ldots, \mathbf{d_n}\}$. As shown in Figure 1, each agent $i$ select its own action-value $Q_i^{a_*}$ and the corresponding action $a_i^*$ according to Equation (9). Because the action space $\mathcal{A}$ is defined as integer ranking scores, that is, each action $a_m$ corresponds to a ranking score. So the ranking scores of all documents can be obtained as $\mathbf{scores} = \{s_1, \ldots, s_n\}$. Finally, a ranked document list $\mathbf{D_{ranked}}$ is obtained based on the descending order of $\mathbf{scores}$.

We show the pseudo-code of ranking process in **Algorithm 1**.

---

**Algorithm 1:** The Ranking Process of MA4DIV

---

1 **Inputs**: Give the query $\mathbf{q}$ and documents $\mathbf{D} = \{\mathbf{d_1}, \ldots, \mathbf{d_n}\}$.
2 **Select Actions**: Each agent $i$ select action-value $Q_i^{a_*}$ to get $\{Q_1^{a_*}, \ldots, Q_i^{a_*}, \ldots, Q_n^{a_*}\}$ based on Equation (9), and the corresponding joint-action $\mathbf{a}^* = \{a_1^*, \ldots, a_i^*, \ldots, a_n^*\}$.
3 **Getting Scores**: Obtain the $\mathbf{scores} = \{s_1, \ldots, s_i, \ldots, s_n\}$ corresponding to the joint-action $\mathbf{a}^*$.
4 **Sorting**: Sort the candidate documents in $\mathbf{D}$ based on $\mathbf{scores}$.
5 **Output**: Get a ranked document list $\mathbf{D_{ranked}}$.

---

## 4.4 Value-Decomposition for MA4DIV

After the ranking process, the reward $\mathcal{R}$ is obtained according to Equation (3) based on $\mathbf{D_{ranked}}$. Obviously, $\mathcal{R}$ is a scalar while there are $n$ action-values $\{Q_1^{a_*}, \ldots, Q_n^{a_*}\}$ for all agents. $\mathcal{R}$ is the evaluation of the global ranking effect, while $Q_i^{a_*}$ is related to the score of a single document. Therefore, we cannot use $\mathcal{R}$ to directly update the action-value functions of all agents.

The MARL algorithms based on value decomposition, such as [27, 32, 35], consider that there is the global utility function $Q_{tot}^*$ can be decomposed into the action-value functions $Q_i^{a_*}$ of all individual agents. $Q_{tot}^*$ represents the action-value utility of the joint-action $\mathbf{a}^* = \{a_1^*, \ldots, a_n^*\}$ under the joint-observation information $\mathbf{o} = \{\boldsymbol{o_1}, \ldots, \boldsymbol{o_n}\}$, and is also a scalar value. The monotonic decomposition relation can be expressed as Equation (10):

$$\arg\max_{\mathbf{a}} Q_{tot}^*(\mathbf{o}, \mathbf{a}) = \begin{pmatrix} \arg\max_{a_1} Q_1(\boldsymbol{o_1}, a_1) \\ \vdots \\ \arg\max_{a_n} Q_n(\boldsymbol{o_n}, a_n) \end{pmatrix} \qquad (10)$$

In online ranking process, all agents adopt a greedy strategy to score documents, which is consistent with the maximization operation on the right side of Equation (10). So we just need to learn how to maximize $Q_{tot}^*$. Since $Q_{tot}^*$ is a global utility function and $\mathcal{R}$ is also used to measure the global ranking effect, we can maximize

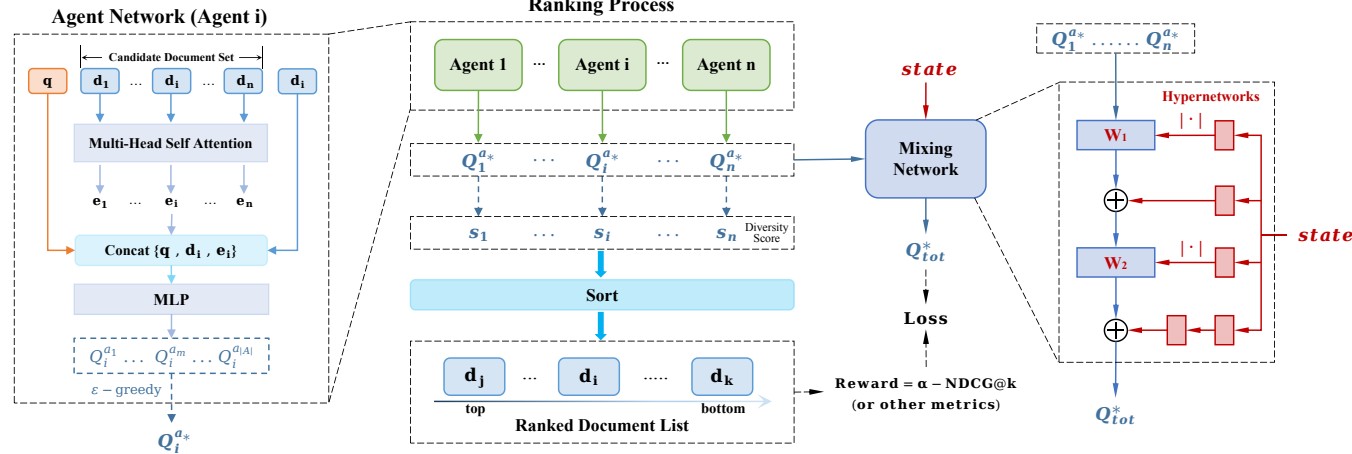

**Figure 1: The Framework of MA4DIV.**

$Q_{tot}^*$ with $\mathcal{R}$, thus maximizing the utility of each individual agent ($Q_i^{a_*}$). The specific loss function will be introduced in the following Section 4.6 (Equation (14)).

Furthermore, QMIX [27] proposes a sufficient but not necessary realization to satisfy Equation (10). Monotonicity can be enforced by a constraint condition on the relationship of partial derivative between $Q_{tot}^*$ and $Q_i^{a_*}$:

$$\frac{\partial Q_{tot}^*}{\partial Q_i^{a_*}} \geq 0, \quad \forall i \in \mathcal{G} \equiv \{1, \ldots, n\} \tag{11}$$

Additionally, in Equation (10) and (11), it is established that $Q_i^{a_*} = \max Q_i(o_i, a_m)$, $a_*^i = \arg\max_{a_m} Q_i(o_i, a_m), \forall m \in \{1, \ldots, |\mathcal{A}|\}$.

### 4.5 Mixing & Hyper Network Structure

To convert the individual action-value functions $Q_i^{a_*}$ into the global utility function $Q_{tot}^*$ and ensure compliance with Equation (11), an architecture consisting of a mixing network and a set of hypernetworks are designed. The right part of Figure 1 illustrates the detail structure.

**Mixing network**, a fully connected network, utilizes the action-values $\{Q_1^{a_*}, \ldots, Q_n^{a_*}\}$ from the agent network as inputs, and then, monotonically combines these inputs to generate $Q_{tot}^*$. The calculation process is shown as:

$$Q_{tot}^* = \mathbf{W}_2 \cdot \text{Elu}\left(\mathbf{W}_1 \cdot [Q_1^{a_*}, \ldots, Q_n^{a_*}] + \mathbf{B}_1\right) + B_2 \tag{12}$$

**Hypernetworks**, several fully connected networks, play a role in generating mixing network weights. Firstly, each hypernetwork takes global state $s$ as input and generate the vectors of $\mathbf{W}_1$, $\mathbf{B}_1$, $\mathbf{W}_2$, $B_2$ respectively. Then, the vectors are reshaped into matrices of appropriate size as parameters for the mixing network. Lastly, to achieve the monotonicity constraint of Equation (11), $\mathbf{W}_1$, $\mathbf{W}_2$ are limited to non-negative by an absolute activation function. However, biases $\mathbf{B}_1$, $B_2$ do not need to be set to non-negative because biases has nothing to do with the partial derivatives between $Q_{tot}^*$ and $Q_i^{a_*}$. The hypernetworks operate as follows:

$$\mathbf{W}_1 = \text{Reshape}\left(\left|\text{FC}_{\mathbf{W}_1}(s)\right|\right), \mathbf{B}_1 = \text{Reshape}\left(\text{FC}_{\mathbf{B}_1}(s)\right)$$
$$\mathbf{W}_2 = \text{Reshape}\left(\left|\text{FC}_{\mathbf{W}_2}(s)\right|\right), B_2 = \text{Reshape}\left(\text{FC}_{\mathbf{B}_2}(s)\right) \tag{13}$$

### 4.6 Training Process of MA4DIV

In this section, we introduce the complete training process for MA4DIV. The parameters of models are $\Theta = \{\theta, \psi, \phi\}$ where $\theta$ represents the parameters of agent network, $\psi$ and $\phi$ are the parameters of mixing network and hypernetworks respectively. The model parameters $\Theta$ is updated on $N$ training data with the form of $\{(q^k, D^k), J^k\}_{k=1}^N$ mentioned in Section 3.2.

---

**Algorithm 2:** The Training Process of MA4DIV

---

1 **Initialize**: The parameters of network $\Theta = \{\theta, \psi, \phi\}$, replay buffer $\mathcal{M}$.

2 **Inputs**: Training data $\{(q^k, D^k), J^k\}_{k=1}^N$.

3 **for** *epoch* = 1 *to* N_epoch **do**

4     // generate data

5     **for** *i* = 1 *to* N **do**

6         Current training data: $\mathbf{q} = q^i$, $\mathbf{D} = D^i$, with subtopic labels $J^i$.

7         Rollout an episode following **Algorithm 1**.

8         Collect a tuple $\mathcal{T} = (\mathbf{o}, \mathbf{s}, \mathbf{a}, \mathcal{R})$ during this episode.

9         Store the tuple $\mathcal{T}$ in $\mathcal{M}$.

10     // update models

11     **for** *update* = 1 *to* N_update **do**

12         Sample a random minibatch $b$ from $\mathcal{M}$.

13         Calculate $y^{tot}$ and loss $\mathcal{L}$ for all sampled data from $b$ based on Equation (15) and (14).

14         Update the parameters of networks $\Theta$ by gradient descent.

15 **Ouput**: A well-trained ranking model with parameters $\Theta$.

---

**Algorithm 2** shows the training pseudo-code. At the beginning, parameters $\Theta$ are initialized randomly and the replay buffer $\mathcal{M}$ is

set to $\varnothing$. A whole training epoch consists of two parts. The first part is the stage of data generation. Given data $\mathbf{q} = q^i$, $\mathbf{D} = D^i$, with subtopic labels $J^i$, tuple $\mathcal{T}$ in the episode is obtained based on the definition in Equation (1, 2, 3). Then, store the tuple $\mathcal{T}$ in replay buffer. Each epoch can generate a kind of permutation of all documents $\{D^k\}_{k=1}^N$ in all training data. The second part is updating models with parameters $\Theta$ for $N\_update$ times. And the update is performed by sampling data through a mini-batch approach from $\mathcal{M}$. The whole network is trained end-to-end by minimizing the Temporal Difference (TD) [33] loss shown in Equation (14) which is widely used in MARL to maximize $Q_{tot}$.

$$\mathcal{L}_{TD} = \sum_{i=1}^{b} \left( y_i^{tot} - Q_{tot}(\mathbf{o}, \mathbf{a}, \mathbf{s}; \Theta) \right)^2 \qquad (14)$$

$$y^{tot} = \mathcal{R} + \gamma \max_{\mathbf{a}'} Q_{tot}(\mathbf{o}', \mathbf{a}', \mathbf{s}'; \Theta') = \mathcal{R} \qquad (15)$$

In Equation (14), $b$ is the batch size of sampled data from replay buffer. In Equation (15), $Q_{tot}(\mathbf{o}', \mathbf{a}', \mathbf{s}'; \Theta')$ is calculated based on information of the next time step, i.e., $\mathbf{o}', \mathbf{a}', \mathbf{s}'$, in typical MDP which contains multiple decision steps. However, we define that MA4DIV only have one time step for an episode, so $\mathbf{o}', \mathbf{a}', \mathbf{s}'$ do not exist and $Q_{tot}(\mathbf{o}', \mathbf{a}', \mathbf{s}'; \Theta')$ equals to 0, that is $y^{tot} = \mathcal{R}$.

## 5 Experiments

Our experiments mainly focus on next research questions:

(**RQ.1**) How does MA4DIV perform on the TREC Web Track datasets when compared to existing diversified search baselines?

(**RQ.2**) How does MA4DIV perform on a larger scale industrial dataset compared to the baselines?

(**RQ.3**) How efficient is MA4DIV in training and inference process?

### 5.1 Datasets and Experimental Settings

We conduct experiments to address above research questions on two datasets, the public TREC 2009~2012 Web Track datasets and the industrial DU-DIV dataset. The details of these two datasets are shown in Table 1.

TREC 2009~2012 Web Track datasets are publicly available and Xia et al. [42] first uses them for training and evaluating diversified search models. Almost all the following works, such as $M^2$Div [11], DALETOR [45], MO4SRD [48], etc., conduct experiments on these datasets. As a widely used test collection for diversified search, the TREC Web track datasets only contains 198 queries, and previous studies often use Doc2vec [18] to obtain the document embeddings on this dataset. In this work, to ensure a fair and meaningful comparison with existing work, we first compare MA4DIV with existing baselines on the this relatively small dataset. Then, we further conduct extensive experiments on a new and larger scale diversity dataset called DU-DIV, which is constructed based on real user data in industrial search engine. More details of datasets can be seen in Appendix D.

We conduct 5-fold cross-validation experiments on the reshaped TREC dataset with the same subset split based on unique queries as in [11]. In the DU-DIV dataset, we randomly sample 80% of 4473 unique queries and their associated document lists to the training set, and the left 20% to the test set. The code framework of MA4DIV

**Table 1: Comparisons of datasets TREC and DU-DIV.**

| Dataset | TREC | DU-DIV |
|---|---|---|
| Source | TREC 2009~2012 Web Track | Real search engine |
| No. of Queries | 198 | 4473 |
| No. of Documents/Query | average 211 docs/query | 15 docs/query |
| No. of Subtopics | maximum 7 subtopics/doc | total 50 subtopics |
| Representation | Doc2vec [18] | BERT [10] |

is based on PyMARL2 framework [15], which is a open-source code framework for multi-agent reinforcement learning. More detail settings of experiments are shown in Appendix E.

We compare MA4DIV with several state-of-the-art baselines in search result diversification, including:

**MMR** [2]: a heuristic approach which select document according to maximal marginal relevance.

**xQuAD** [28]: a representative approach that explicitly models different aspects underlying the query in the form of subqueries.

**MDP-DIV** [42]: a single-agent reinforcement learning approach which model the diverse ranking process as MDP.

**$M^2$Div** [11]: a single-agent reinforcement learning approach which utilizes Monte Carlo Tree Search to enhance ranking policy.

**DALETOR** [45]: a method for approximating evaluation metrics that optimizes an approximate and differentiable objective function.

**MO4SRD** [48]: another method to make evaluation metrics differentiable and optimize the approximating evaluation metrics.

### 5.2 Experimental Results

Table 2 and 3 report the performances of all baselines and our MA4DIV in terms of the six diversity performance metrics, including $\alpha$-NDCG@5, $\alpha$-NDCG@10, ERR-IA@5, ERR-IA@10, S-recall@5, S-recall@10 on the TREC web track datasets and the DU-DIV dataset respectively. The detail of evaluation metrics can be found in Appendix B. We highlight the top three algorithms on each metric by darkening the background color of the numbers.

*5.2.1 The Results on TREC web track datasets.* In Table 2, $M^2$Div and MO4SRD have the best performance in term of $\alpha$-NDCG, our MA4DIV performs worse than these two algorithms but better than the others. Moreover, MA4DIV takes the lead in both ERR-IA and S-recall metrics, with MO4SRD also exhibiting strong performance in S-recall metric. Considering multiple evaluation metrics comprehensively, MA4DIV, $M^2$Div and MO4SRD are the best three algorithms on the TREC web track datasets. (answer to **RQ.1**)

As mentioned in Table 1, TREC web track datasets have only 198 unique queries, deep learning methods are prone to overfitting on this dataset. Figure 2 shows the training curve of MA4DIV on the TREC web track datasets. We can see, after about the 11 iterations, MA4DIV achieves the best performance on the test data, and the subsequent iterations fall into overfitting the relatively small training set. However, the evaluation metric $\alpha$-NDCG@10 on the training set grows to more than 0.8. These results indicate that: (1) Our MA4DIV is capable to effectively optimize the diversity metric $\alpha$-NDCG@10. (2) It could easily overfit the relatively small training set of the TREC web track dataset, which only contains 198 unique queries. (3) MA4DIV has the potential to perform better in the search result diversification task when given a larger dataset.

**Table 2: Performance comparison of baselines and MA4DIV on TREC web track datasets. The best result in each metric is bold and the top-3 results are shaded. "*" indicates the difference between the baseline and M4DIV (ours) is statistically significant.**

| Method | $\alpha$-NDCG@5 | $\alpha$-NDCG@10 | ERR-IA@5 | ERR-IA@10 | S-recall@5 | S-recall@10 |
|--------|-----------------|------------------|----------|-----------|------------|-------------|
| MMR | 0.4273* | 0.5059* | 0.2015* | 0.2153* | 0.6058* | 0.7848* |
| xQuAD | 0.4451* | 0.5296* | 0.2108* | 0.2243* | 0.5994* | 0.7887* |
| MDP-DIV | 0.4987* | 0.5663* | 0.2662 | 0.2865 | 0.6310* | 0.7853* |
| M$^2$Div | **0.5144*** | 0.5798* | 0.2338* | 0.2448* | 0.6593 | 0.8028 |
| DALETOR | 0.5008 | 0.5703 | 0.2237* | 0.2355* | 0.6451* | 0.8012 |
| MO4SRD | 0.5135* | **0.5815*** | 0.2280* | 0.2396* | 0.6576 | **0.8074** |
| MA4DIV (ours) | 0.5056 | 0.5724 | **0.2680** | **0.2882** | **0.6600** | 0.8070 |

**Table 3: Performance comparison of baselines and MA4DIV on DU-DIV dataset. The best result in each metric is bold and the top-3 results are shaded. "*" indicates the difference between the baseline and M4DIV (ours) is statistically significant.**

| Method | $\alpha$-NDCG@5 | $\alpha$-NDCG@10 | ERR-IA@5 | ERR-IA@10 | S-recall@5 | S-recall@10 |
|--------|-----------------|------------------|----------|-----------|------------|-------------|
| MMR | 0.7416* | 0.8171* | 0.7583* | 0.8100* | 0.6062* | 0.8481* |
| xQuAD | 0.7075* | 0.8277* | 0.7745* | 0.8201* | 0.5670* | 0.8520* |
| MDP-DIV | 0.8101* | 0.8623* | 0.7694* | 0.8204* | 0.6389* | 0.8590* |
| M$^2$Div | 0.8089* | 0.8610* | 0.7633* | 0.8142* | 0.6468* | 0.8608* |
| DALETOR | 0.7981* | 0.8543* | 0.7450* | 0.7955* | 0.6447* | 0.8550* |
| MO4SRD | 0.7989* | 0.8553* | 0.7467* | 0.7969* | 0.6432* | 0.8556* |
| MA4DIV (ours) | **0.8187** | **0.8702** | **0.7836** | **0.8342** | **0.6534** | **0.8699** |

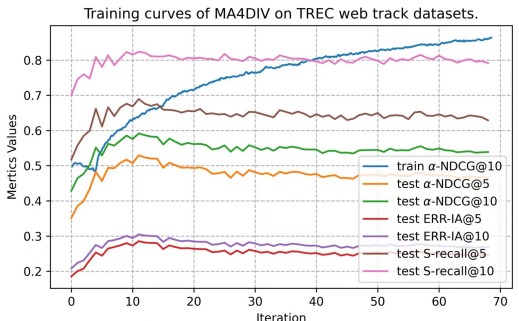

**Figure 2: Training curves of MA4DIV on TREC web track datasets.**

Therefore, we construct a larger diversity dataset called DU-DIV from a real industrial search engine. And then we show the experiments on DU-DIV dataset.

*5.2.2 The Results on DU-DIV dataset.* Table 3 shows results in different evaluation metrics on the DU-DIV dataset. MA4DIV achieves state-of-the-art performances on all six evaluation metrics. Heuristic algorithms, MMR and xQuAD, perform poorly on $\alpha$-NDCG. Comparing to single-agent reinforcement learning methods, MDP-DIV and M$^2$Div, MA4DIV has a better performance on evaluation metrics which demonstrate that MA4DIV can improve the diversity of search results. MA4DIV also outperforms DALETOR and MO4SRD which optimize the approximate value of the evaluation metrics. This verifies that directly optimizing the evaluation metrics is indeed better than optimizing the approximation of the evaluation metrics. All these results demonstrate that the proposed MA4DIV

method outperforms existing search result diversification methods on an industrial scale dataset (answering **RQ.2**).

Interestingly, we notice that MDP-DIV, which does not perform well on the TREC web track datasets, is the second best performing algorithm on the DU-DIV dataset, following our MA4DIV in terms of $\alpha$-NDCG@5 and $\alpha$-NDCG@10. In MDP-DIV, the reward function is defined as: $R(s_t, a_t) = \alpha\text{-}DCG[t+1] - \alpha\text{-}DCG[t]$, where $\alpha\text{-}DCG[t]$ is the discounted cumulative gain at the $t$-th ranking position. And the objective function of MDP-DIV is to maximize the cumulative rewards in one episode, that is $G_t = \sum_{k=0}^{n-1-t} \gamma^k R_{t+k+1}$. Consider that Xia et al. [42] sets $\gamma=1$ and $\alpha\text{-}DCG[t]=0$, the objective function can be rewritten as $G_t = \alpha\text{-}DCG[n]$, which is different from $\alpha\text{-}NDCG$ with just a normalization factor. In addition, the objective function of our MA4DIV defined in Equation (5) is equal to $\alpha\text{-}NDCG$. Similar to MA4DIV, MDP-DIV also directly optimizes one of the end evaluation metrics in SRD. Therefore, this provides a theoretical analysis for MDP-DIV to be able to outperform other baselines.

## 5.3 Comparison and Analysis of Efficiency

*5.3.1 Training Time on TREC web track datasets.* To highlight the efficiency of our MA4DIV, we also compare several algorithms from the perspective of training time. Table 4 shows the comparison of training time taken by different algorithms to achieve their optimal performance on the TREC web track datasets. To ensure a fair comparison, all the algorithms in Table 4 run on the same machine.

We can see that M$^2$Div takes over a day to reach optimal performance. This is because M$^2$Div utilizes Monte Carlo Tree Search for policy enhancement, which is highly time-consuming. The time taken by MDP-DIV and MO4SRD is close and significantly less than M$^2$Div. MA4DIV takes the shortest time, only 20∼25 minutes. This is due to the efficient exploration ability of using a multi-agent to

813
814
815
816
817
818
819
820
821
822
823
824
825
826
827
828
829
830
831
832
833
834
835
836
837
838
839
840
841
842
843
844
845
846
847
848
849
850
851
852
853
854
855
856
857
858
859
860
861
862
863
864
865
866
867
868
869
870

**Table 4: The comparison of the training time taken by different algorithms to achieve the optimal value of metrics.**

| Method | Training Time |
|--------|--------------|
| MDP-DIV | 5~6 hours |
| MO4SRD | 6~7 hours |
| $M^2Div$ | 1 day + |
| MA4DIV (ours) | 20~25 mins |

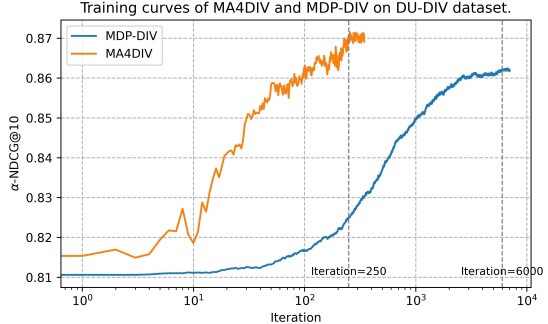

**Figure 3: Training curves of MA4DIV and MDP-DIV on DU-DIV dataset.**

model the ranking process, in which a permutation of the given documents set ($n$ documents) can be obtained at one time step. The single agent reinforcement learning algorithm MDP-DIV can only explore a permutation of the given documents after a whole episode which contains $n$ time steps. This is the main reason why the training time of MA4DIV is significantly shorter than MDP-DIV. In MO4SRD, the complexity of calculating all approximate ranking positions $r_i$ in $\alpha$-DCG is $O(n^2)$ and the complexity of calculating all $c_{li}$ approximations is $O(n^2m)$, which are time-consuming parts. While MA4DIV directly optimizes $\alpha$-NDCG without complex approximations, which takes significantly less time.

In summary, the training time of MA4DIV is significantly shorter than that of other baselines, which demonstrate its training efficiency and answers **RQ.3**.

*5.3.2 Iteration Times on DU-DIV dataset.* As the optimization goals of MA4DIV and MDP-DIV are essentially the same (we discussed in the analysis of the results on DU-DIV dataset), we are interested in comparing the learning efficiency of the proposed multi-agent RL approach and the existing single-agent RL method.

Figure 3 shows the training curves of MA4DIV and MDP-DIV on the DU-DIV dataset. We conduct the experiments of these two algorithms with the same learning rate which is set to $1×10^{-5}$. From the curves, we can see that, after 250 iterations, $\alpha$-NDCG of MA4DIV basically converges to about 0.87. However, $\alpha$-NDCG of MDP-DIV still fails to reach the optimal value at around 6000 iterations. This indicates that MA4DIV has higher exploration and exploitation efficiency compared with MDP-DIV, which also answers **RQ.3**.

The main reason for the results just mentioned is as follows: MDP-DIV adopts an on-policy RL method to optimize the cumulative rewards. A feature of on-policy RL is that the currently sampled data is discarded after it is utilized. While MA4DIV is a off-policy

**Table 5: The inference complexity of different diversified search algorithms. $n$ is the number of candidate documents.**

| Method | Inference Complexity |
|--------|---------------------|
| Heuristic Approaches | $O(n)$ |
| Single-Agent RL Approaches | $O(n)$ |
| Approximate Metric Approaches | $O(1)$ |
| MA4DIV (ours) | $O(1)$ |

RL method. From 9-th line and 12-th line of **Algorithm 2**, we can see that the sampled data is stored in replay buffer $\mathcal{M}$, then update the model multiple times in minibatch approach. In addition, each data stored in replay buffer $\mathcal{M}$ may be sampled multiple times to update the ranking model. Therefore, the data utilization efficiency of MA4DIV is higher than that of MDP-DIV, which is the main reason why MA4DIV converges significantly faster.

*5.3.3 Analysis of Inference Complexity.* Table 5 shows the inference complexity of different approaches. We can see that the Heuristic Approaches, including MMR [2] and xQuAD [28], have to perform inference process in $O(n)$. And $n$ is the number of candidate documents. In addition, Single-Agent RL Approaches, such as MDP-DIV [42], also has a $O(n)$ inference complexity. These two approaches can only select one document in one time step, so the inference complexity is equal to the number of candidate documents.

Because DALETOR [45] and MO4SRD [48] optimize the approximation of evaluation metrics and can score for all candidate documents at once, these approaches have $O(1)$ inference complexity. In our MA4DIV, the ranking scores $\{s_1, \ldots, s_n\}$ can be obtained in one time step, so MA4DIV also has the lowest inference complexity with $O(1)$.

The Section 5.3 shows that MA4DIV have the shorter training time on the TREC web track datasets, the faster convergence rate on the DU-DIV dataset and the lowest inference complexity. And these advantages jointly demonstrate that MA4DIV has high efficiency in training and inference process, which answers **RQ.3**.

## 6 Conclusions and Future Work

In this paper we have proposed a novel method for search result diversification using MARL, called MA4DIV. To alleviate the sub-optimal problem and improve the efficiency of training, we consider each document as an agent and model the diversity ranking process as a multi-agent cooperative task. We conduct experiments on the TREC web track dataset to prove the effectiveness and potential of our MA4DIV. Furthermore, we construct a larger scale dataset called DU-DIV using the real data from industrial search engine. And MA4DIV achieve state-of-the-art performance and high learning efficiency on the DU-DIV dataset.

A major contribution of this paper is to introduce a multi-agent reinforcement learning framework for the ranking optimization task. We argue that modeling ranking process in the multi-agent setting can be used not only to optimize the diversity of search results, but also to optimize other ranking tasks, such as relevance ranking and fairness ranking. Therefore, in future work, we will try to propose a general ranking framework base on multi-agent reinforcement learning and apply it to a variety of ranking tasks.

871
872
873
874
875
876
877
878
879
880
881
882
883
884
885
886
887
888
889
890
891
892
893
894
895
896
897
898
899
900
901
902
903
904
905
906
907
908
909
910
911
912
913
914
915
916
917
918
919
920
921
922
923
924
925
926
927
928

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

# Appendix

## A  Why do existing methods lead to suboptimal diversity ranking?

### A.1  Why the "greedy selection" approach leads to suboptimality.

In the task of Search Result Diversification, the "greedy selection" methods [2, 11, 28, 42] typically refer to selecting the document that can provide the most additional information at each step when generating a diversified result list, i.e., making a locally optimal choice. However, this method is prone to suboptimal rankings for the following reasons:

- **Local Optimality Instead of Global Optimality:** The greedy selection method chooses the document most beneficial to the current ranking at each step, but this method only considers the optimal solution for the current step, not the global optimality of the entire ranking list. Hence, although each step is a locally optimal choice, the final result may not be the best overall ranking.
- **Lack of Holistic Perspective:** As the greedy selection is performed step by step, it does not take into account all possible combinations of documents and their overall effects. This might result in missing out on document combinations that could provide broader topic coverage.
- **Ignoring Interactions Among Documents:** Greedy selection often does not fully consider the interrelationships among documents. For instance, some documents might be highly related in content, and their consecutive appearance might reduce user satisfaction and the diversity of the results.
- **Mismatch Between Objective Function and Evaluation Metrics:** The objective function used in the training of the greedy selection method may not match the metrics used in the final evaluation. This inconsistency could lead to optimising for a target in the training process that does not match the performance needing evaluation.
- **Lack of Flexibility:** Greedy selection methods are usually not flexible; once the ranking order is determined, it is difficult to adjust. If the selected documents are not optimal, i.e., there is an error in the ranking compared to the optimal one, then the documents chosen based on the currently selected documents will result in cumulative errors.

### A.2  Why optimizing an approximation of the objective function leads to suboptimality.

Methods of optimizing the approximation of the objective function [45, 48] can often achieve parallel inference (faster than the serial selection of "greedy selection"), but may result in suboptimal ranking results due to the following issues:

- **Mismatch Between Objective Function and Evaluation Metrics:** There is a mismatch between the optimization objective used in the training process and the evaluation metrics used in the final assessment, which may prevent the model from accurately learning to achieve the best diversification effect.
- **Low Exploration Efficiency:** Such methods are less capable of exploring different document ranking combinations. When faced with the challenge of exploring as many different document ranking combinations as possible in a vast combination space, it leads to an increase in the time complexity of training and makes it difficult to find the global optimum.

## B  Diversity Evaluation Metrics

In this section, we introduce three diversity evaluation metrics, namely $\alpha$-NDCG [7], ERR-IA [4] and S-recall [49]. The basic setting is that there are $n$ documents in a query, and each document may cover 1 to $m$ subtopics.

### B.1  $\alpha$-NDCG

Firstly, the $\alpha$ discounted cumulative gain ($\alpha$-DCG) is defined as Equation (16).

$$\alpha\text{-}DCG = \sum_{i=1}^{n} \sum_{l=1}^{m} \frac{y_{il}(1-\alpha)^{c_{li}}}{log_2(1+r_i)} \quad (16)$$

$y_{il} = 1$ means that subtopic $l$ is covered by document $i$ and $y_{il} = 0$ is the opposite. $\alpha$ is a parameter between 0 to 1, which quantifies the probability of a reader getting the information about a specific subtopic from a relevant document. $r_i$ is the ranking position of document $i$, and $c_{li}$ is the number of times that the subtopic $l$ being covered by the documents on the prior positions to $r_i$. Specifically, $c_{li}$ is defined as $c_{li} = \sum_{j:r_j \leq r_i} y_{jl}$.

Then, dividing the $\alpha$-DCG of a given documents list by the $\alpha$-$DCG_{ideal}$ of an ideal ranking list yields the normalized $\alpha$-NDCG,

$$\alpha\text{-}NDCG = \frac{\alpha\text{-}DCG}{\alpha\text{-}DCG_{ideal}} \quad (17)$$

The metric of top $k$ documents in a given list, called $\alpha-NDCG@k$, is usually used to measure the diversity of the documents list.

### B.2  ERR-IA

The ERR-IA is defined in Equation (18).

$$ERR\text{-}IA = \sum_{i=1}^{n} \frac{1}{r_i} \sum_{l=1}^{m} \frac{1}{m} \left( \prod_{j:r_j \leq r_i} \left( 1 - \frac{2^{y_{jl}} - 1}{2^{y_l^{max}}} \right) \right) \frac{2^{y_{jl}} - 1}{2^{y_l^{max}}} \quad (18)$$

If $y_{il}$ is a binary label ($y_{il} = 0$ or 1), the Equation (18) can be rewritten as Equation (19).

$$ERR\text{-}IA = \sum_{i=1}^{n} \frac{1}{r_i} \sum_{l=1}^{m} \frac{1}{m} \frac{y_{il}}{2^{c_{li}+1}} \quad (19)$$

Similar to $\alpha$-NDCG@k, ERR-IA@k is also often used in practice.

### B.3  S-recall

The S-recall is always defined on top $k$ documents of a ranked list, i.e.,

$$S\text{-}recall@k = \frac{\left| \cup_{i=1}^{k} subtopics(d_i) \right|}{\left| \cup_{i=1}^{n} subtopics(d_i) \right|} \quad (20)$$

Obviously, S-recall@k represents the proportion of subtopics number recalled in the first top $k$ documents to the number of subtopics contained in the whole document list.

## C  The Detail of MHSA

### C.1  The structure of MHSA

The MHSA module is an essential component in Transformer [34] and its variants. Given an input matrix $\mathbf{X} \in \mathbb{R}^{n \times L}$, where $n$ and $L$ denote the number of documents in candidate document set $\mathbf{D}$ and the embedding vector dimension of $\mathbf{d_i}$. MHSA computes a weighted sum of all the input tokens for each token. The weight (also called attention score) between two tokens is calculated using their similarity.

Specifically, MHSA first linearly projects the input matrix into query $\mathbf{Q}$, key $\mathbf{K}$, and value $\mathbf{V}$ matrices, i.e., $\mathbf{Q} = \mathbf{XW}_Q$, $\mathbf{K} = \mathbf{XW}_K$, and $\mathbf{V} = \mathbf{XW}_V$, where $\mathbf{W}_Q$, $\mathbf{W}_K$, and $\mathbf{W}_V$ are learnable weight matrices with dimensions $L \times d_k$, $L \times d_k$, and $L \times d_v$ respectively, and $d_k$ and $d_v$ are the dimensions of keys/values. Then, the attention score between each two tokens is computed as $\mathbf{A_s} = \text{softmax}(\frac{\mathbf{QK}^T}{\sqrt{d_k}})$.

The output of MHSA is $\mathbf{O} = \mathbf{A_s V}$, with $\mathbf{O} \in \mathbb{R}^{n \times d_v}$.

MHSA further enhances the model capacity by applying the above process multiple times in parallel, which is called multi-head mechanism. Suppose there are $h$ heads, the output after multi-head is $\mathbf{O} = [\mathbf{O}_1; \mathbf{O}_2; \cdots ; \mathbf{O}_h]\mathbf{W}_O$, where $[\cdot]$ denotes concatenation, and $\mathbf{O}_i$ and $\mathbf{W}_O$ are the output of the $i$-th head and a learnable weight matrix with dimensions $h \cdot d_v \times d$, respectively.

### C.2  The reason for using MHSA

Why do we employ the MHSA module to obtain $\mathbf{e_i}$ rather than using other networks like MLP to compute the cross features of all documents directly? The main reason is as follows:

According to [22], the ranking model should satisfy the permutation invariance property which is described as following Definition:

DEFINITION 1 (PERMUTATION INVARIANCE).
*The target ranking for a given set is the same regardless of the order of documents in the set. In other words, no matter how exchange the positions of $d_i$ and $d_j$ in inputs information, it will not affect the final ranking result.*

A method to construct a ranking model in accordance with Permutation Invariance involves initially attributing scores to the documents via a permutation equivariant scoring function, followed by ordering based on these scores. Pang et al. [22] also proved that the multi-head attention block is permutation equivariant. Therefore, the MHSA module is used to compute higher-level cross feature vectors between documents.

## D  The Introduction of Datasets

We conduct experiments to address above research questions on two datasets, TREC 2009~2012 Web Track datasets and DU-DIV dataset. TREC 2009~2012 Web Track datasets are publicly available and Xia et al. [42] first uses them for training and evaluating diversified search models. Almost all the following works, such as M$^2$Div [11], DALETOR [45], MO4SRD [48], etc., conduct experiments on these datasets. As a widely used test collection for diversified search, the TREC Web track dataset only contains 198 queries, and previous studies often use Doc2vec [18] to obtain the document embeddings on this dataset. In this work, to ensure a fair and meaningful comparison with existing work, we first compare

MA4DIV with existing baselines on the this relatively small dataset. Then, we further conduct extensive experiments on a new and larger scale diversity dataset called DU-DIV, which is constructed based on real user data in industrial search engine. The Figure 4 shows the average number of subtopics of the 15 documents with ideal ranking permutation in the DU-DIV dataset.

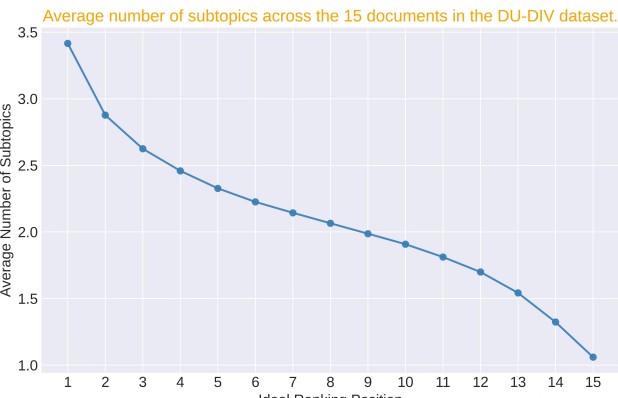

Figure 4: Average number of subtopics across the 15 documents in the DU-DIV dataset. The abscissa from 1 to 15 represents positions 1 to 15 in the ideal permutation, and the ordinate is the average of the number of subtopics contained in the documents of the corresponding position.

More details of these two datasets are shown in Table 6. The DU-DIV dataset has 4473 queries, far more than the TREC datasets. The documents in the DU-DIV dataset are the top-15 documents retrieved by the industrial search engine for each query, which makes this dataset particularly useful for the diversified ranking task in which we need to rerank these top documents for diversity. In the TREC web track datasets, each document contains a maximum of 7 subtopics. In the DU-DIV dataset, we predefined 50 subtopics, such as law, healthcare, education, etc. Each document corresponds to one or more of the predefined subtopics. Moreover, while previous studies on the TREC web track dataset often use the Doc2vec model to represent queries and documents in vector with $L$=100 dimensions, the DU-DIV dataset utilizes the more powerful model BERT [10] to encode queries and documents into vectors with $L$=1024 dimensions.

## E  Experimental Settings

Considering that TREC datasets contain only 198 queries, this is prone to overfitting for existing deep learning models. In addition, the average number of documents related to each query is about 211, but a large number of these documents do not contain any subtopics. Therefore, we reshape TREC dataset. That is, from each of the original document lists in TREC (198 lists in total, with an average of 211 documents per query), we subsample 30 documents at a time to form a new document list. This process runs multiple times over a document list associated with the same query, resulting in a reshaped dataset with a total of 6,232 document lists (198 unique queries). And it's crucial to ensure that an appropriate number of the 30 documents sampled contain at least one subtopic. Figure 5 shows

**Table 6: Comparisons of datasets TREC and DU-DIV.**

| Dataset | TREC | DU-DIV |
|---|---|---|
| Source | TREC 2009~2012 Web Track | Real search engine |
| No. of Queries | 198 | 4473 |
| No. of Documents per Query | average 211 docs/query | 15 docs/query |
| No. of Subtopics | maximum 7 subtopics/doc | total 50 subtopics |
| Representation | Doc2vec [18] | BERT [10] |

**Table 7: Hyperparemeters of MA4DIV on different dataset.**

| Dataset | $|\mathcal{A}|$ | $L$ | $H$ | $z$ |
|---|---|---|---|---|
| TREC | 30 | 1 | 4 | 64 |
| DU-DIV | 15 | 1 | 4 | 256 |

the number of documents with subtopics in these 6,232 document lists. We can see that there are 1,053, 1,717, and 1,980 document lists with $16 \sim 20$, $21 \sim 25$, $26 \sim 30$ documents containing any subtopics, and 758 document lists with $1 \sim 5$ documents containing any subtopics. This means that after our sampling, most document lists contain a rich number of subtopics, and some document lists have a small number of subtopics. Overall, this ensures that the data is comprehensive.

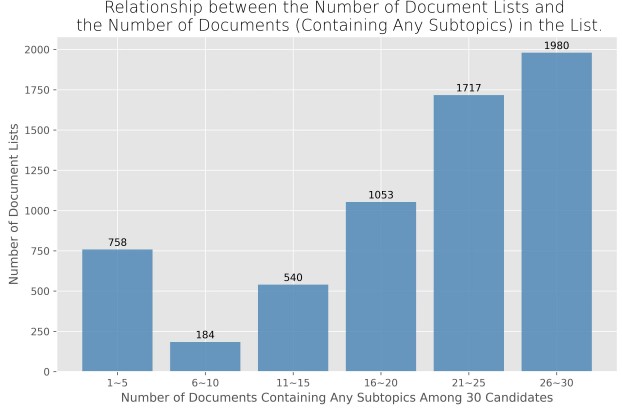

Figure 5: This figure illustrates the relationship between the quantity of newly sampled document lists and the count of documents that include any subtopics present in the sampled list.

We conduct 5-fold cross-validation experiments on the reshaped TREC dataset with the same subset split based on unique queries as in [11]. In the DU-DIV dataset, we randomly sample 80% of 4473 unique queries and their associated document lists to the training set, and the left 20% to the test set.

The code framework of MA4DIV is based on PyMARL2 framework [15], which is a open-source code framework for multi-agent reinforcement learning. The key hyperparameters of MA4DIV on two datasets are shown in Table 7. $|\mathcal{A}|$ is the dimension of action space, i.e. the number of discrete scores $(1, \ldots, |\mathcal{A}|)$. $L$, $H$ and $z$ respectively represent the number of MHSA block, attention heads and attention embedding dimensions of MHSA.

We reproduce the code of DALETOR and MO4SRD, and ensure that these two algorithms have the same parameter scale of MHSA structure as our MA4DIV in Table 7, that is, the $L = 1$ and $H = 4$, $z$ is 64 and 256 respectively on two datasets.

