# OpenReview forum: "MA4DIV: Multi-Agent Reinforcement Learning for Search Result Diversification"
_ACM.org/TheWebConf/2025/Conference — WWW 2025 Oral_

### Official Review · Reviewer_PNxU · 2024-11-30

**Novelty:** 6
**Technical Quality:** 6

**Review:**

This work proposes a novel method called MA4DIV, which uses multi-agent reinforcement learning (MARL) to improve search result diversification (SRD). SRD aims to ensure that search results cover a broad range of subtopics to satisfy users with diverse information needs. The paper argues that existing SRD methods suffer from limitations in effectiveness and efficiency.

Originality and Significance:
The use of MARL for SRD is a novel approach. MA4DIV models each document as an independent agent in a cooperative multi-agent setting, where agents work together to maximize a shared reward function based on diversity metrics. This approach allows for direct optimization of diversity metrics like α-NDCG, leading to potentially more effective and efficient SRD.

Quality and Clarity:
The paper is well-written and presents a clear and detailed description of the MA4DIV framework. The authors thoroughly explain the rationale behind their design choices, such as using a multi-head self-attention (MHSA) module to capture cross-document features and employing value decomposition to optimize global diversity.

Pros:
1. Improved Effectiveness: MA4DIV achieves state-of-the-art performance on both the public TREC web track datasets and a larger-scale industrial dataset (DU-DIV) across multiple diversity metrics. This suggests that directly optimizing diversity metrics through MARL leads to more effective SRD compared to existing methods.
2. Enhanced Efficiency: MA4DIV demonstrates significant efficiency gains in both training and inference. Its training time is considerably shorter than baselines, and it converges faster to optimal solutions. This efficiency stems from the multi-agent approach, allowing simultaneous exploration of multiple ranking permutations.
3. Direct Optimization of Diversity Metrics: MA4DIV optimizes the actual diversity metrics during training, avoiding approximations that could lead to suboptimal results.
4. Permutation Invariance: By utilizing MHSA, MA4DIV ensures permutation invariance, meaning the ranking results are not affected by the initial order of documents. This property aligns with the desired behavior of a ranking model.

Cons:
1. Potential Scalability Issues: As the number of documents increases, the computational complexity of MARL can grow considerably. This could limit the scalability of MA4DIV to very large document sets.
2. Hyperparameter Sensitivity: MARL algorithms often involve numerous hyperparameters that can significantly influence performance. Fine-tuning these hyperparameters might require extensive experimentation.

**Questions:**

Scalability of MA4DIV for Large Document Sets: The authors mention that the computational complexity of MARL can increase significantly with a larger number of documents. Could the authors elaborate on the scalability of MA4DIV for large document sets, specifically addressing:
1. How does the training time and computational resources required by MA4DIV scale with the number of documents?
2. Are there any strategies or modifications to the framework that could mitigate potential scalability issues?

**Reviewer Confidence:**

3: The reviewer is confident but not certain that the evaluation is correct

**Scope:**

3: The work is somewhat relevant to the Web and to the track, and is of narrow interest to a sub-community

---

### Official Review · Reviewer_Z6Hh · 2024-12-02

**Novelty:** 6
**Technical Quality:** 4

**Review:**

This work builds upon previous studies like MDP-DIV and M2Div, which employed single-agent reinforcement learning, by extending them to a multi-agent reinforcement learning framework for the task of search result diversification (SRD).

## Quality
- The idea of using multi-agent reinforcement learning for SRD is novel and intriguing.
- However, several recent (but not too recent) related works are omitted. These works appear to be highly relevant and are evaluated on the same datasets, which weakens the claims made in the paper by not including them in the discussion. For example:
  - [Passage-aware Search Result Diversification](https://dl.acm.org/doi/abs/10.1145/3653672)
  - [Multi-grained Document Modeling for Search Result Diversification](https://dl.acm.org/doi/abs/10.1145/3652852)
  - [Search Result Diversification Using Query Aspects as Bottlenecks](https://dl.acm.org/doi/10.1145/3583780.3615050)

- While the authors provide arguments regarding the sampling approaches on TREC in the appendix, it remains unclear why they did not use a larger sampling size to better simulate real-life systems. Sampling only 30 documents per query (or 15 documents on DU-DIV) seems unrealistic and raises concerns about potential biases in the approach.

## Clarity
The presentation is of high quality. The ideas and experiments are clearly articulated and easy to follow.

## Originality
The ideas presented are creative and show originality.

## Significance
It is difficult to assess the significance at this stage.

**Questions:**

1. Could you explain why recent works were not included in the discussion? How do these approaches compare to your proposed method?
2. What are the limitations or challenges that prevented the use of a larger sampling size (e.g., more than 30 documents per query) to better simulate real-life systems? Could a larger sampling size alter the conclusions drawn from your experiments?

**Reviewer Confidence:**

4: The reviewer is certain that the evaluation is correct and very familiar with the relevant literature

**Scope:**

3: The work is somewhat relevant to the Web and to the track, and is of narrow interest to a sub-community

---

### Official Review · Reviewer_mAij · 2024-12-06

**Novelty:** 3
**Technical Quality:** 3

**Review:**

This paper introduces a multi-agent reinforcement learning (MARL) framework for search result diversification (SRD), demonstrating strong methodological rigor and originality. While generally well-written and structured, clarity in technical details could be enhanced. The study improves over baselines in efficiency and effectiveness, addressing a critical challenge in information retrieval and hinting at broader applications in ranking tasks. The use of cooperative MARL for directly optimizing diversity metrics without approximations represents a novel aspect in the field.

### Pros
1. Novel application of MARL to SRD, demonstrating state-of-the-art performance.
2. Direct optimization of diversity metrics (e.g., 𝛼-NDCG) without approximations.
3. Significant improvements in training and inference efficiency, as evidenced by experiments.
4. Comprehensive evaluation on public and large-scale industrial datasets.

### Cons
1. Limited discussion on scalability to datasets with extremely large numbers of documents.
2. Insufficient detail on hyperparameter tuning and specific configurations for reproducibility.
3. The cooperative dynamics among agents lack detailed analysis, potentially underexploring inter-agent behavior optimization.
4. Practical deployment challenges in real-world search engines are not explicitly addressed.

**Questions:**

1. How does MA4DIV handle datasets with an extremely large number of candidate documents, and are there any constraints or performance bottlenecks in such scenarios?

2. Can you provide more details on how agent cooperation is modeled and optimized in the multi-agent reinforcement learning framework?

3. What specific hyperparameters and configurations were used during training, and are there any plans to release the code or additional details to enhance reproducibility? The source code of the StepAgent method should be provided for validation during the review period.

4. What are the practical challenges you foresee in deploying MA4DIV in real-world search engines, and how might they be mitigated?

5. It can be observed from Table 2 that MA4DIV is inferior to M^2Div and MO4SDR on small TREC web track datasets. The author should explain the rationale behind it.

**Reviewer Confidence:**

3: The reviewer is confident but not certain that the evaluation is correct

**Scope:**

2: The connection to the Web is incidental, e.g., use of Web data or API